# Multi-Temporal InSAR Deformation Monitoring Zongling Landslide Group in Guizhou Province Based on the Adaptive Network Method

**Yu Zhu** [1], **Bangsen Tian** [1,*], **Chou Xie** [1,2], **Yihong Guo** [1], **Haoran Fang** [1,3], **Ying Yang** [1,3], **Qianqian Wang** [4], **Ming Zhang** [1,3], **Chaoyong Shen** [5] **and Ronghao Wei** [6]

1   Aerospace Information Research Institute, Chinese Academy of Sciences, Beijing 100094, China
2   Laboratory of Target Microwave Properties, Deqing Academy of Satellite Applications, Hangzhou 313200, China
3   University of Chinese Academy of Sciences, Chinese Academy of Sciences, Beijing 100049, China
4   School of Earth Sciences and Resources, China University of Geosciences, Beijing, 100083, China
5   The Third Surveying and Mapping Institute of Guizhou Province, Guiyang 550004, China
6   Zhejiang Institute of Hydraulics & Estuary (Zhejiang Institute of Marine Planning and Design), Hangzhou 310017, China
*   Correspondence: tianbs@aircas.ac.cn

**Abstract:** Due to the influence of atmospheric phase delays and terrain fluctuation in complex mountainous areas, traditional PS-InSAR technology often fails to select enough measurement points (MPs) and loses effective MPs during phase unwrapping. To solve this problem, this paper proposes an adaptive network construction algorithm, which combines the permanent scatterer (PS) points with the distributed scatterer (DS) points. Firstly, to ensure the extraction quality of the DS points, the covariance matrix of DS points is estimated robustly. Secondly, based on the traditional Delaunay triangulation network, an adaptive network construction method is proposed, which can adaptively increase edge redundancy and network connectivity by considering the edge length, edge coherence, edge number, and spatial distribution. Finally, a total of 31 RADARSAT-2 SAR images that cover the Zongling landslide group in Guizhou Province were used to prove the effectiveness of proposed method. The results show that the quantity of available DS points can be increased by 23.6%, through the robust estimation of the covariance matrix. In addition, it is demonstrated that the proposed network construction algorithm can balance the number, distribution, and quality of edges in the dense and sparse areas of MPs adaptively. This adaptive network construction approach can maintain good connectivity and avoid losing effective MPs to the greatest extent, especially when the scattering points are far away from the reference points. In short, the proposed algorithm improves the number of effective MPs and accuracy of phase unwrapping.

**Keywords:** PS-InSAR; adaptive network; covariance matrix; robust estimation

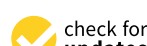



## 1. Introduction

Persistent scatterer interferometric synthetic aperture radar (PS-InSAR) has been proven to be a powerful geodetic technique for measuring the deformations of the earth's surface in space and time by using a stack of synthetic aperture radar (SAR) images [1–7]. However, PSs often correspond to point-wise radar targets with highly stable backscattering behavior, which results in few measurement points (MPs) in non-urban areas, such as desert land and vegetated mountain areas [8]. In addition, effective MPs in non-urban areas are vulnerable to the influence of atmospheric phase delays, resulting in phase unwrapping errors and the failure in extracting the displacement information [9].

In view of the above problems, PS-InSAR technology has been improved by two methods. One method is to develop the distributed scatterer (DS) to increase the density of coherent target points. DS points usually correspond to the pixels whose neighboring

pixels share similar reflectivity values, such as bare soil, sparsely vegetated land, or desert areas [10]. Under the Gaussian scattering assumption [11], the statistics of DSs can be described by a zero-mean, multivariate, complex Gaussian probability distribution function [12]. Typical algorithms that use DS point targets include SBAS [13], StaMPS-SBAS [14], TCP-InSAR [15], and QPSS [16]. Particularly, SqueeSAR [10], which was proposed in 2011, has become the most popular method for increasing the number and density of MPs by combining PS and DS. The key steps of processing DS include: (1) statistically homogeneous pixels (SHPs) selection, (2) coherence matrix estimation, and (3) phase triangulation optimization. The results of phase triangulation optimization are influenced by the accuracy of the statistically homogeneous pixel selection and coherent matrix estimation. The Kolmogorov–Smirnov (KS) test [10] and Anderson–Darling (AD) test [17] are the common methods for selecting SHPs, but they only use the information of amplitude value and only work for stacks of at least eight images [18]. Subsequently, under the assumption of stability in spatial statistics or temporal statistics, Cao [19] and Schmitt et al. [20] pre-estimated the covariance matrix in the spatial or temporal domain with a small window and then used the generalized likelihood ratio of covariance matrix to select SHPs. However, since the scattering characteristics of the ground objects are subject to temporal changes (such as vegetation growth, soil moisture, freeze-thaw state changes, and so on) and nonstationary of spatial distribution, e.g., not all the pixels in the small local window share similar statistical properties, the pre-estimated covariance matrix may be not accurate, thus affecting the accuracy of the coherence matrix estimation. Thus, how to reduce the impact of heterogeneous points, which are inevitably in the process of SHP selection, on the estimation of coherence matrix is a key point.

The other method for improving PS-InSAR technology is network optimization for MPs. The key point is how to form a stable reference network to connect these MPs and then monitor the deformation over larger areas based on the use of spatial difference (SD) [13]. Obviously, the closer the pixels are, the lower the atmospheric impact and the higher the final estimation accuracy that can be obtained by the spatial difference of the pixel pairs. The development of network construction methods can be divided into three stages: the first proposed method is the single reference network that is constructed by the spatial difference between all pixels and the reference point, and it is only applicable for MPs within several hundred meters from the reference point. When the MPs are far from the reference point, the error will exceed the limit and, thus, cannot be phase unwrapped correctly. Subsequently, the approach of transmitting network construction is proposed, which combines the local and global reference points to increase the MPs in a way of regional growth. However, the results of this process may be not reliable because the error is also accumulated during the transmission process. Currently, redundant network construction is the most popular method. This method generates connected graphs with many redundant edges from adjacent point pairs, which makes it possible to obtain a solution that is more robust for outliers and noise. Under the framework of redundant network, the following improvements have been proposed: (1) The Delaunay triangulation network (DTN) [21] is usually used to connect all sparse points [22]. However, it is independent of the phase continuity rule, due to the fact that it just checks whether the convex hull of two neighboring triangles contain other points and whether the triangles overlap. This implies that many edges may have higher phase gradient in a spatial network when deformation signals, atmospheric turbulence, and phase noise are presented in the interferogram. (2) Considering the temporal coherence to construct the network, based on DTN, it is beneficial to improve the result of phase unwrapping by using temporal coherence to select edges with high quality [9,23], but this may result in isolated networks, due to the fact that the connectivity of the DTN is damaged. (3) The network is built using the connectivity or distance threshold method [24,25], and the point pairs within a certain threshold are linked to form redundant networks. Since each point pair is connected by the same connectivity or distance threshold, if the threshold is small, some connected regions far apart cannot be connected to form a lot of "isolated island" or multi-connected regions.

If the threshold is set too large, there will be too many redundant edges in the area with the dense point, which increases the computational cost and, thus, lowers the efficiency of phase unwrapping. Zhang-Feng MA et al. [26] recently enhanced the DTN by considering the distance threshold and time coherence coefficient. However, this method cannot balance the number of edges in dense regions and sparse regions of the measurement point. As a result, it is very significant to construct an edge adaptive redundant network, while ensuring the accuracy of phase unwrapping.

In order to accurately estimate the covariance matrix of DS points by reducing the impact of noise points in the statistically homogeneous pixel selection process and construct an adaptive network with global distribution to achieve accurate phase unwrapping, we propose an adaptive network algorithm for MPs, which jointly processes the PS and DS points. The validation experiment is conducted in the Zongling landslide group (Nayong, Guizhou, China).

The rest of this paper is organized as follows. Section 2 introduces the study area and data source. Section 3 briefly describes the basic process of multi-temporal InSAR technology and then presents the specific solutions to the above two issues, i.e., robust estimation of the covariance matrix for DS and adaptive network construction. The C-band RADARSAT-2 data were used to verify the Zongling landslide group and prove the effectiveness of the proposed method, and the results and discussion are listed in Section 4. Finally, this paper is concluded in Section 5.

## 2. Study Area and Data Source

The study area is located in at the southeast foothills of the Wumeng Mountain, northeast of Zongling Town, Nayong County, and in the northwest of Guizhou province (Figure 1a). It is in the slope zone of Yunnan-Guizhou Plateau, the second step of the transition between the first step of the Qinghai-Tibet Plateau and the third step of the eastern hilly plain. The geotectonic unit is part of the northeast-trending tectonic deformation zone of the Zunyi fault arch Bijie in the Qianbei platform uplift of the Yangtze Para platform [27]. Due to the extremely fragile karst geological environment, heavy rainfall, and mining activities in the karst areas of Nayong Country [28–33], landslides on mountainous areas frequently happen, thus resulting in a significant loss of life and property. There are a series of historical landslide activities in these study areas, including the Zhongling landslide, the Zuojiaying landslide, and so on (Figure 1b). The study area and the elevation information are shown in Figure 1a. The Google Earth optical images and historical landslides in the study area are shown in Figure 1b.

In this paper, 31 RADARSAT-2 images collected from 19 April 2017 to 3 April 2020 with an ultra-wide fine mode were used to validate the proposed method. The spatial resolution was 5 m, the coverage was 125 km by 125 km, and the revisit period was 24 days. Furthermore, the one arc-second shuttle radar topography mission (SRTM) digital elevation model (DEM) with a resolution of 30 m was used to remove the topographic phase and geocoding.

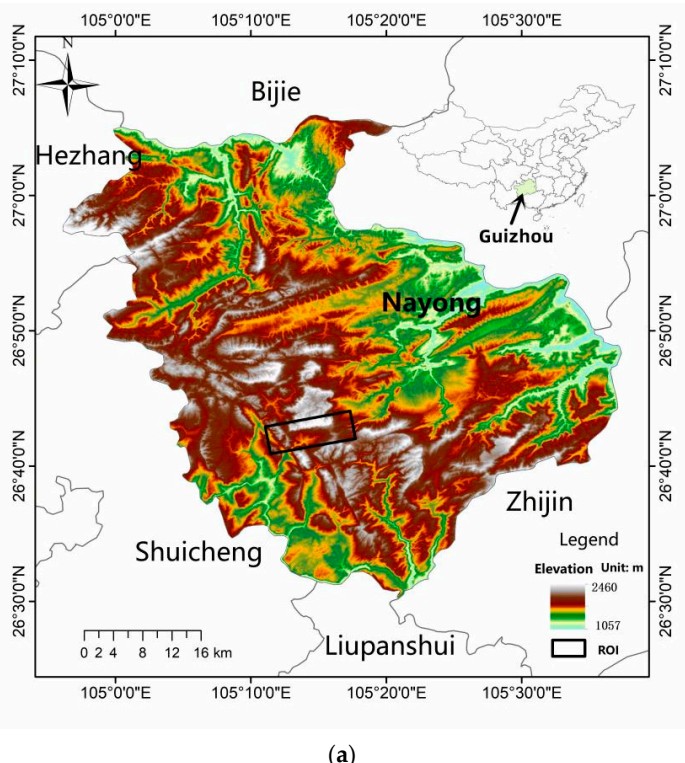

(**a**)

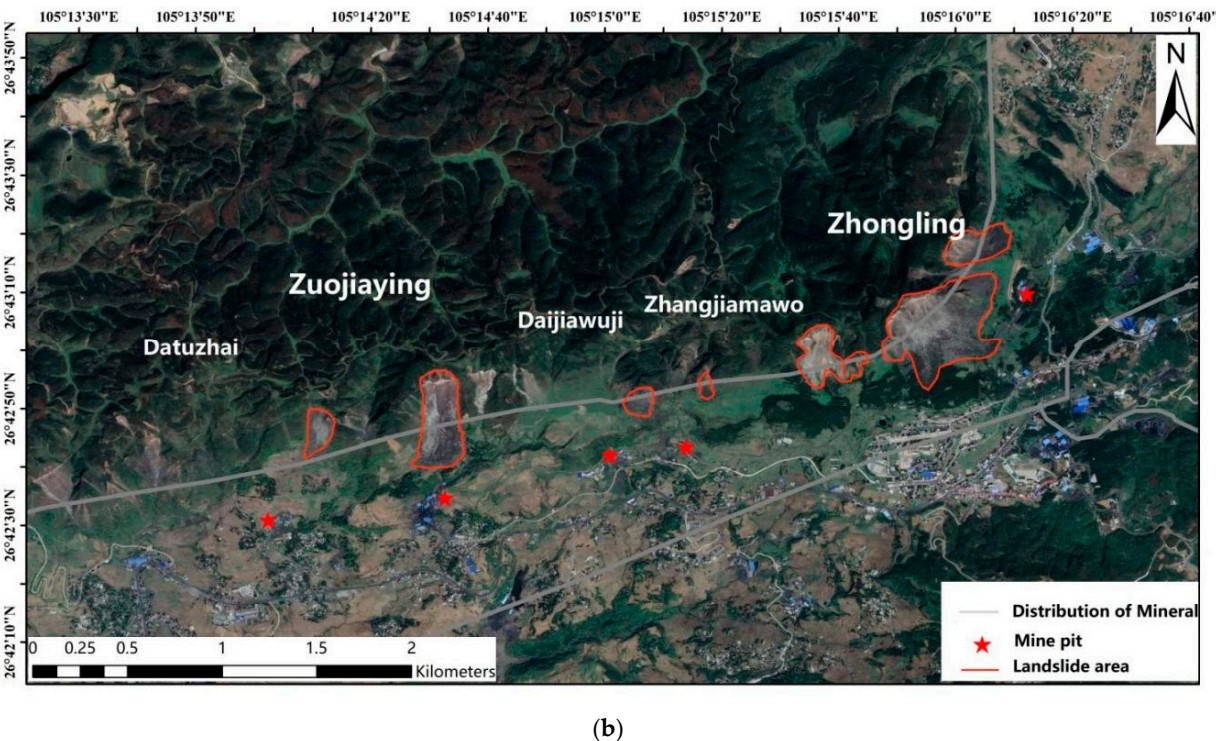

(**b**)

**Figure 1.** (**a**): The elevation distribution map of the study area at a resolution of 30 m. The black frame represents the region of interest that covered the Zongling landslide group. (**b**): The optical images of the study area from Google Earth. The red curve represents the coverage of the historical landslide area, while the area surrounded by gray lines represents the distribution of coal mines, and the red stars represent the mine pit.

## 3. Method

As shown in Figure 2, the processing flow of multi-temporal InSAR mainly consists of four steps: (1) Data pre-processing, including co-registration, region of interest (ROI) clipping, interferogram generation, and topographic phase removal; (2) PS and DS selections, including extraction of high-quality DS points through robust estimation of the covariance matrix and selection of PS points through the amplitude dispersion method and spectral diversity method; (3) Network construction, including constructing the initial Delaunay triangulation, choosing the seed points and seed edges, calculating the seed point spatial distribution, intensifying and expanding the network edges, and selecting and optimizing the skeleton network; (4) Post-processing, including calculating the linear deformation rate and residual elevation, residual phase unwrapping, estimating the atmospheric phase, and geocoding. In this section, the proposed methods for improving point selection and network construction are introduced in detail.

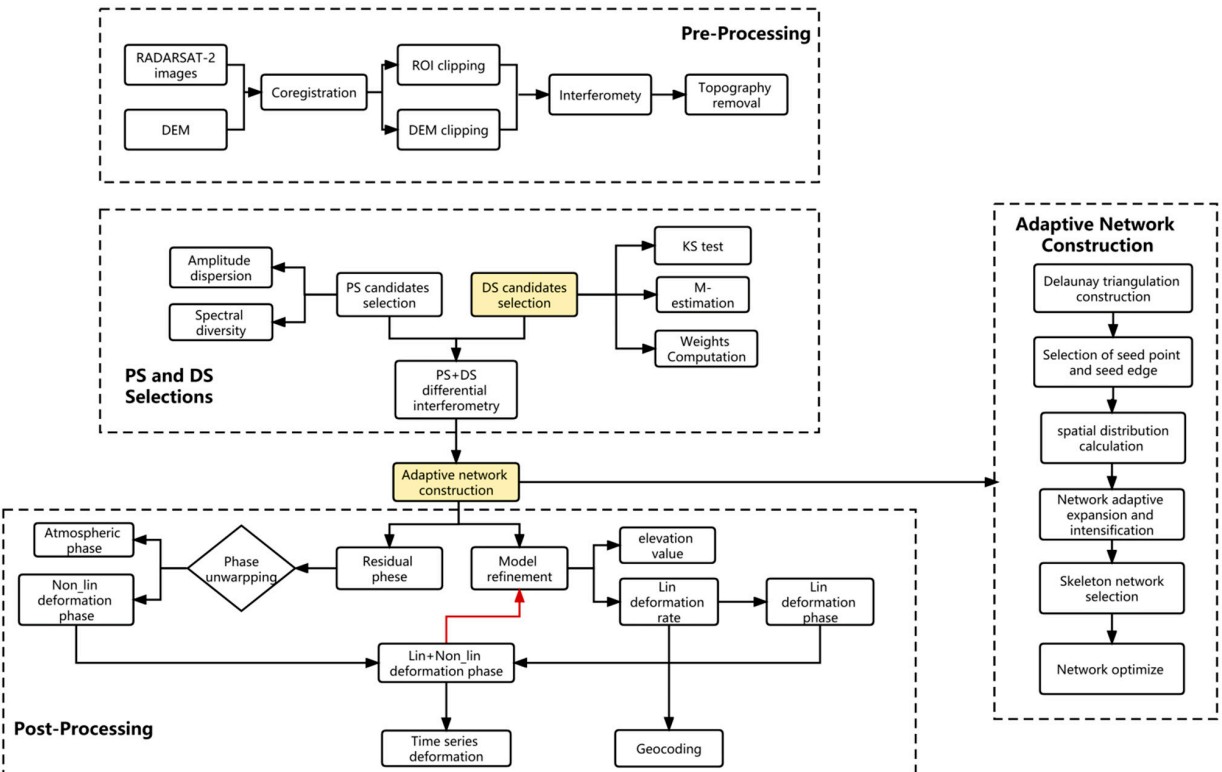

**Figure 2.** The main flow chart of data processing, in which the dashed boxes represent the four steps of data pre-processing, PS and DS selections, adaptive network construction, and post-processing. This paper mainly improves DS point selection and adaptive network construction, as shown in the yellow background.

### 3.1. Robust Estimation of DS Points

Based on the central limit theorem, we made the assumption that the SAR data vector g follows a complex multivariate circular Gaussian (CCG) distribution. The probability density function is given by:

$$f_g(g) = \frac{1}{\pi^N \det(C)} \exp\left(-g^H C^{-1} g\right) \tag{1}$$

where C is the covariance matrix, H indicates Hermitian conjugation, det( ) denotes the determinant, and N is the dimension of data vector g. Given a set of M spatial station-

ary samples, $G = [g_1, g_2, \ldots, g_M]$, and the maximum likelihood estimation (MLE) of the covariance matrix can be obtained as:

$$C_{MLE} = \frac{1}{M} \sum_{m=1}^{M} g_m g_m^H \tag{2}$$

However, with the increasing of SAR resolution, clutter environments are most often heterogeneous, and using the sample covariance matrix of the heterogeneous samples results in significant performance degradation of DS selection. In order to mitigate the deleterious effects of heterogeneous samples, the robust weighted M-estimation was applied in this study, which is expressed as follows [34,35]:

$$C_{k+1} = \frac{1}{M} \sum_{m=1}^{M} w(x_m(C_k)) g_m g_m^H \tag{3}$$

where $w(\cdot)$ denotes the equivalent weight function, and the general equivalent weight function $w(\cdot)$ can be expressed as:

$$w(x) = \frac{-\partial \ln f(x)}{x \partial x} \tag{4}$$

To derive a meaningful weighting function for some heavily tailed distributions, the non-Gaussian SAR pixel distribution is modeled as a complex t-distribution [36,37]:

$$f_g(g) = \frac{\tau(v+N)}{(\pi v)^N \tau(v) \det(C)^N} \left(1 + \frac{1}{v} g^H C^{-1} g\right)^{-(v+N)} \tag{5}$$

where $\tau(\cdot)$ is a gamma function, and $v$ is the degree of freedom (DoF). The complex t-distribution is a realistic model for SAR observations, as the DoF allows for a quantitative definition of its deviation from CCG. This distribution approaches CCG as the DoF approaches $+\infty$, and it becomes more heavily tailed as the DoF decreases toward zero. In practical terms, this provides flexibility for handling different levels of outliers.

By substituting formula (5) into formula (4), and considering the transformation from complex t-distribution to real-valued t-distribution, the corresponding weighting function can be derived thereafter as follows:

$$w(x) = \frac{2N + v}{v + 2x^2} \tag{6}$$

To best safeguard against unknown heavily tailed distributions, $v$ can be set to 0, so the weighting function $w(x)$ will be expressed as

$$w(x) = \frac{N}{g_m^H C^{-1} g_m} \tag{7}$$

Further, to remove the iterative process, by assuming that C equals $\bar{I} \cdot E$, where $\bar{I}$ is the expected intensity of g points and E is the identity matrix, the weighting function can be changed to:

$$w(x) = \bar{I} \cdot N \cdot ||g_m||^{-2} \tag{8}$$

Finally, the robust covariance matrix of any point can be estimated by the weighted homogeneous point strength, that is:

$$C_{SCM} = \frac{N \bar{I}}{M} \sum_{m=1}^{M} ||g_m||^{-2} g_m g_m^H \tag{9}$$

This method is referred to as a symbolic covariance matrix (SCM) [38,39], where only the direction of each multivariate sample is considered. SCM does not require iteration and is robust against outliers.

### 3.2. Adaptive Network

Given a series of coregistered SAR images, let us consider the kth interferogram phase between the ith and jth image. Let $\delta\phi_{a,\,k}$ be the phase difference between two points identified by the edge a connecting them. Let $\delta h_a$ and $\delta v_a$ be the height (precisely, the residual height if a DEM was used to flatten the phase) and velocity differences between the same two points. Moreover, let $B_k$ and $T_k$ be the vertical baseline and time baseline between the ith and jth images, respectively. The phase $\delta\phi_{a,\,k}$ can be modeled as:

$$\delta\phi_{a,k} = \frac{4\pi}{\lambda}[T_k\delta v_a + aB_k\delta h_a + \varepsilon_{a,k}]_{2\pi} \tag{10}$$

where $\lambda$ is the wavelength, $\varepsilon_{a,k}$ includes mainly thermal and speckle noise, and $[\ ]_{2\pi}$ indicates modulo 2 operation. The noise $\varepsilon_{a,k}$ is by definition small for closest MPs, and (10) is a fundamental equation for constructing network and retrieving the residual height $\delta h_a$ and velocity differences $\delta v_a$ associated with the edge a. A useful parameter is the temporal coherence associated with the edge a, defined as:

$$\gamma_a = \max_{v_a,h_a}\left|\sum_k w_{a,k}e^{j\varepsilon_{a,k}}\right| \tag{11}$$

where $w_{a,k}$ are possible weights chosen, according to a given criterion (and they can depend on the amplitude values of the acquired SAR images). The temporal coherence $\gamma_a$ can be used to measure the quality of an edge, and an edge can be retained if $\gamma_a$ is greater than a given threshold.

In fact, given a set of N points, (N-1)N possible edges exist. However, it is not necessary to test for all possible edges. It is crucial to determine how to employ a limited number of edges with high quality to connect as many points as feasible. So, we propose an adaptive construct network approach, as follows.

### 3.2.1. Initial Delaunay Triangulation Network Construction

The initial network is realized through DTN. The advantage of the DNT is to connect adjacent point pairs with short edges as much as possible, while ensuring that all points are connected.

### 3.2.2. Selection of Seed Point and Seed Edge

The coherence of each edge in the initial network is calculated according to the formula (11), and the low-quality edge is removed according to the coherence threshold. Meanwhile, these isolated candidate MPs are temporarily masked off, while the candidate MPs associated with high-quality edges are retained. This procedure is repeated with the retained candidate MPs until the new DTN remains constant. For these retained points, each point is connected with at least one high-quality edge, and the retained points and high-quality edges are called seed points and seed edges.

### 3.2.3. Adaptive Expansion and Intensification of Network

During the iterative construction process of DTN, the spatial distribution knowledge of the MPs in the adjacent regions of seed points were obtained by refreshing the edge set connected by seed points. Specifically, the edge set connected by any seed point i retained in the previous step was statistically analyzed, and the edge direction was divided into k quadrants (e.g., eight quadrants in this study), and the longest edge in each quadrant was counted. According to the construction rules of DTN, a longer edge indicates that there are fewer adjacent scattering points in this direction for seed point i. Therefore, we used the edge length $R^{\wedge}k\_t = \partial\cdot\min(R^{\wedge}k\_max, R\_c)$ as the threshold to connect all the possible MPs in quadrant k to achieve adaptive intensification of the network, where $\partial$ was the adjustable scaling factor. In practice, if $R^{\wedge}i\_max = 0$, $R^{\wedge}i\_t = \partial\cdot R\_c$, where $R\_c$ is the upper bound of the edge length. This procedure was repeated for all seed points,

and all possible candidate edge sets were determined. After the temporal coherence of the candidate edges were calculated, the low-quality edges could be eliminated from the adaptive intensification network, according to a threshold of temporal coherence, and then isolated points were removed, too. Finally, the set of primary points and edges were determined.

### 3.2.4. Selection of the Skeleton Network and Network Optimization

Based on the primary points and edges, the coherence threshold T0 and connectivity threshold C0 can be adjusted to make sure whole-scene distribution with the high-quality skeleton network. Optionally, based on the skeleton network, the rest points are connected to the skeleton by setting a lower coherence threshold T1 and connectivity threshold C1, and the final connected network with global distribution is defined.

## 4. Experimental Results

### 4.1. RADARSAT-2 Pre-Processing

SAR image pre-processing included the selection of the reference image, SLC image registration, and clipping of the registered image, according to the ROI [40]. Meanwhile, some basic interferometric processing steps were also performed by the DIFF and GEO module using the Gamma software (version 2016), such as multi-looking, intensity map generation, and DEM map generation and geocoding.

In this study, 31 acquisitions of RADARSAT-2 data were pre-processed. The image obtained on 12 August 2018 was selected as the reference image to register the SAR data of the study area based on the distribution of the time baseline and spatial baseline [41]. Then, the short time baseline threshold was set to 120 days to ensure the connectivity of the time-series of interferograms. Finally, we obtained 100 interferograms, and the results are presented in Figure 3.

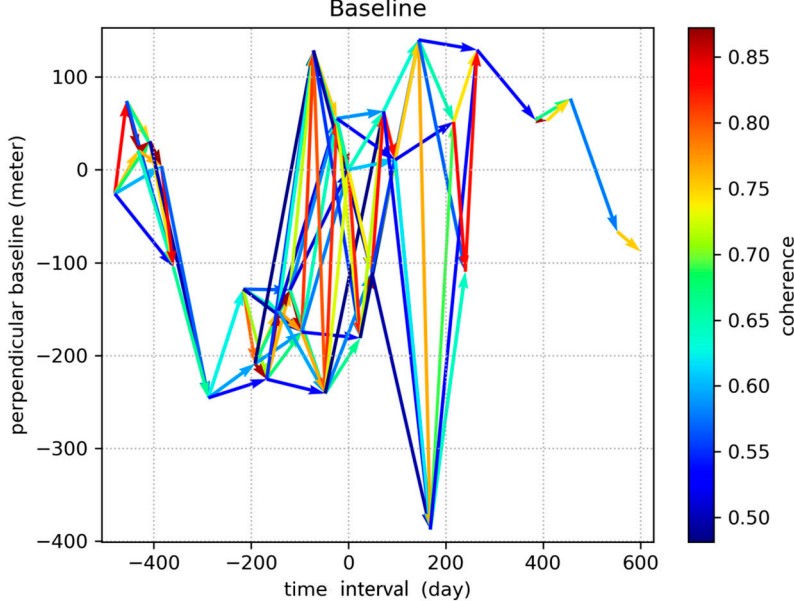

**Figure 3.** Spatial and temporal baseline distribution of the selected interferograms.

### 4.2. PS and DS Selections

The amplitude dispersion method and spectral diversity method were employed to select PS points [42]. Consider the distribution function of SAR intensity image with a long tail [43], the KS test method was used to select homogeneous pixels, and the covariance matrix of the DS point was robustly estimated through the process described above. In this study, the search window of homogeneous pixels was set to $15 \times 15$ px, and the center pixel of the window was used as a reference point to identify homogeneous pixels. Once

these homogeneous pixels were selected, the covariance matrix could be achieved by robust estimation, and finally, the phase-link could be performed by the maximum likelihood method.

The DS points of the entire study area were selected through no robust (maximum likelihood estimation) and robust estimation, respectively, and these corresponding results are compared in Figure 4. Using only no robust estimation, 84,985 valid pixels were obtained. Robust estimation is able to increase its number to 130,104, which implies an improvement of 53.1%. This better performance of robust estimation is due to the fact that it can reduce the weight of heterogeneous samples, thus improving the coherence, which results in an increase of the candidate measurement point density.

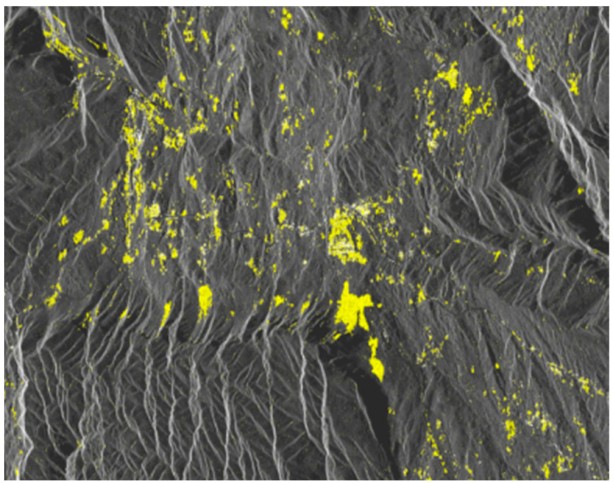
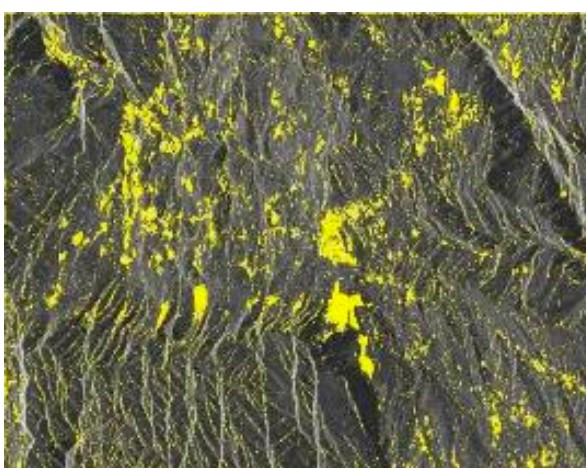

(**a**) DS points before robust estimation
(**b**) DS points after robust estimation

**Figure 4.** The number of DS points before/after robust estimation.

### 4.3. Adaptive Network Construction

The adaptive network experiment was conducted following the preceding steps. The initial edge length and coherence of DTN are shown in Figure 5a,b, respectively. The edge length and coherence of the seed point network after DTN iteration are shown in Figure 5c,d. Compared with Figure 5a,b, the edge length of the seed points were shorter, and the coherence was improved. In addition, the number of edges increased from 6043 to 40,706. Figure 5e,f shows the primary point edge length and coherence in the network after spatial adaptive intensification. As against Figure 5c,d the area with fewer edges in the space was adaptively intensified based on the seed points. The number of edges increased further to 367,652. Finally, the skeleton network was selected and optimized based on the primary points and edges, and the number of edges decreased to 214,113. Figure 5g shows the edge length of the optimized skeleton network, where the maximum edge length was 999.843 m and the average edge length was 74.95 m. Compared with the initial DNT, where the maximum edge length was 3679 m and the average edge length was 999.63 m, the edge length quality was improved. Figure 5h shows the coherence of the optimized skeleton network, with an average of 0.75. Therefore, the adaptive optimization network maintains good connectivity and coherence with a small number of edges.

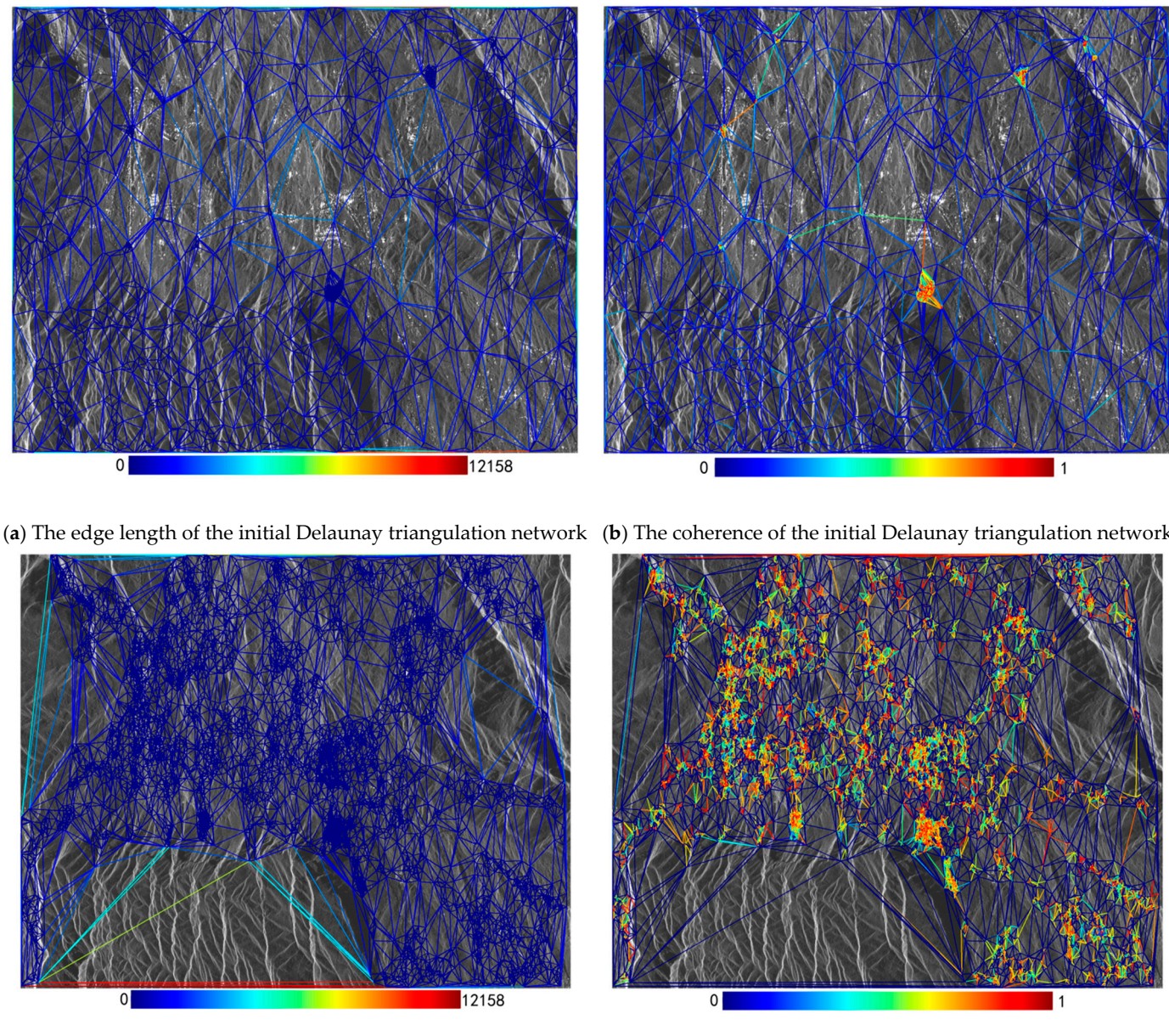

(**a**) The edge length of the initial Delaunay triangulation network  (**b**) The coherence of the initial Delaunay triangulation network

(**c**) The edge length of the seed point network

(**d**) The coherence of the seed points network

**Figure 5.** *Cont.*

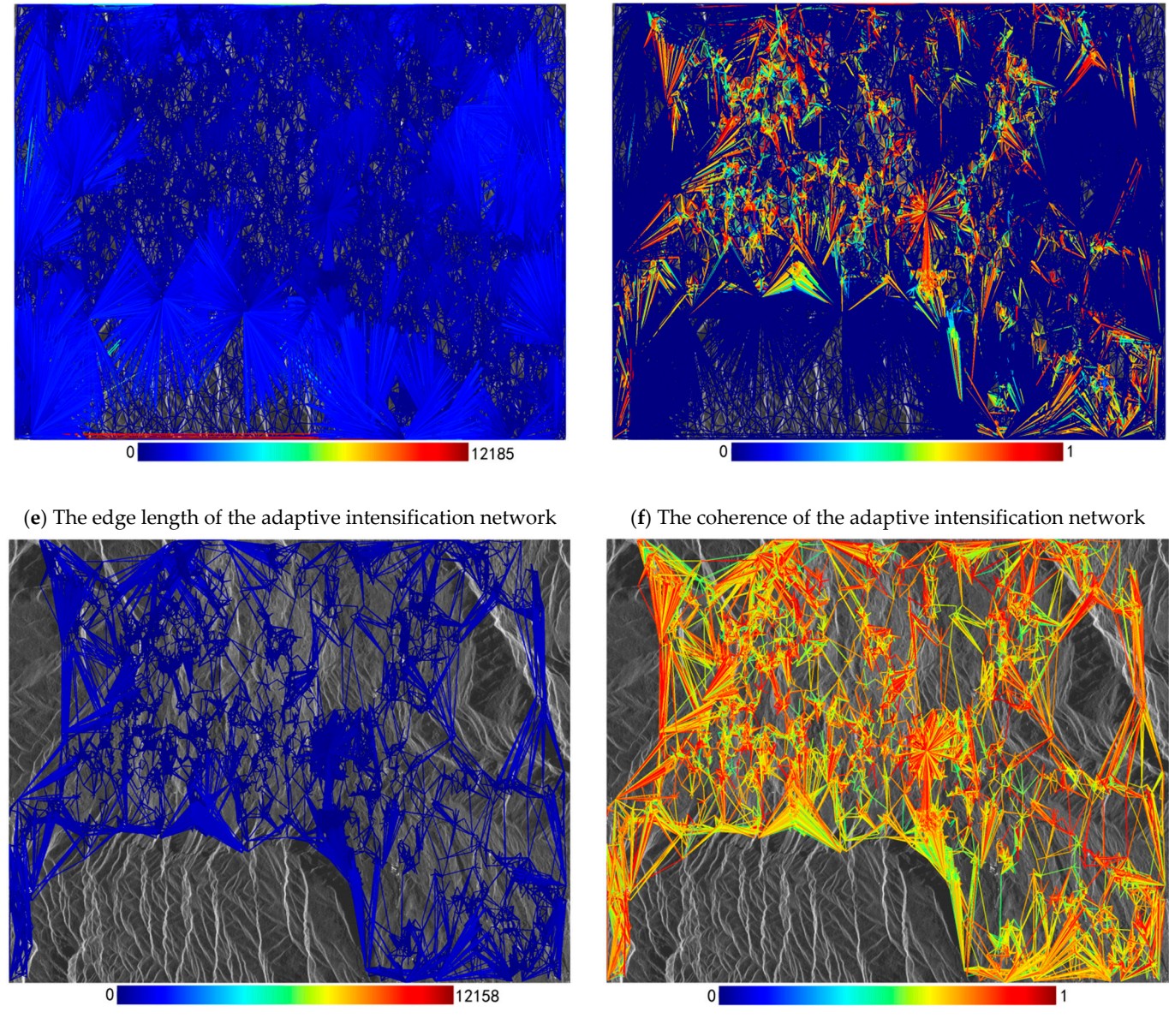

(**e**) The edge length of the adaptive intensification network

(**f**) The coherence of the adaptive intensification network

(**g**) The edge length of the adaptive optimization network

(**h**) The coherence of the adaptive optimization network

**Figure 5.** The whole process of adaptive network construction, the color bars in the (**a**,**c**,**e**,**g**) refer to the edge length, and the color bars in the (**b**,**d**,**f**,**h**) refer to the edge coherence.

*4.4. Post-Processing*

Post-processing included residual elevation phase compensation, residual phase unwrapping, atmospheric phase estimation, and deformation rate calculation. First, the linear deformation rate and residual elevation phase component were subtracted from the original differential interference phase, and then the residual phase of each interferogram was unwrapped using the minimum cost flow method to obtain the unwrapped residual phase. Subsequently, the unwrapped residual phase was filtered in the spatial-temporal domain to obtain the atmospheric phase. Finally, the nonlinear deformation phase was obtained by removing the atmospheric phase from the unwrapped residual phase, and the total deformation phase was obtained by adding the linear deformation phase and the nonlinear deformation phase. Figure 6 shows a post-processed deformation rate map of the study area. The effective MPs cover the whole study area. The vast majority of region deformation in the figure was small (green points), and a few areas with abnormal deformation were

distributed on the landslide body. The region covering red points represents a negative deformation rate for earth surface decline where is landslide's upper edge, and the blue points represent a positive deformation rate for earth surface ascent, which is the landslide's lower edge. The result is consistent with the identified landslide area and tendency.

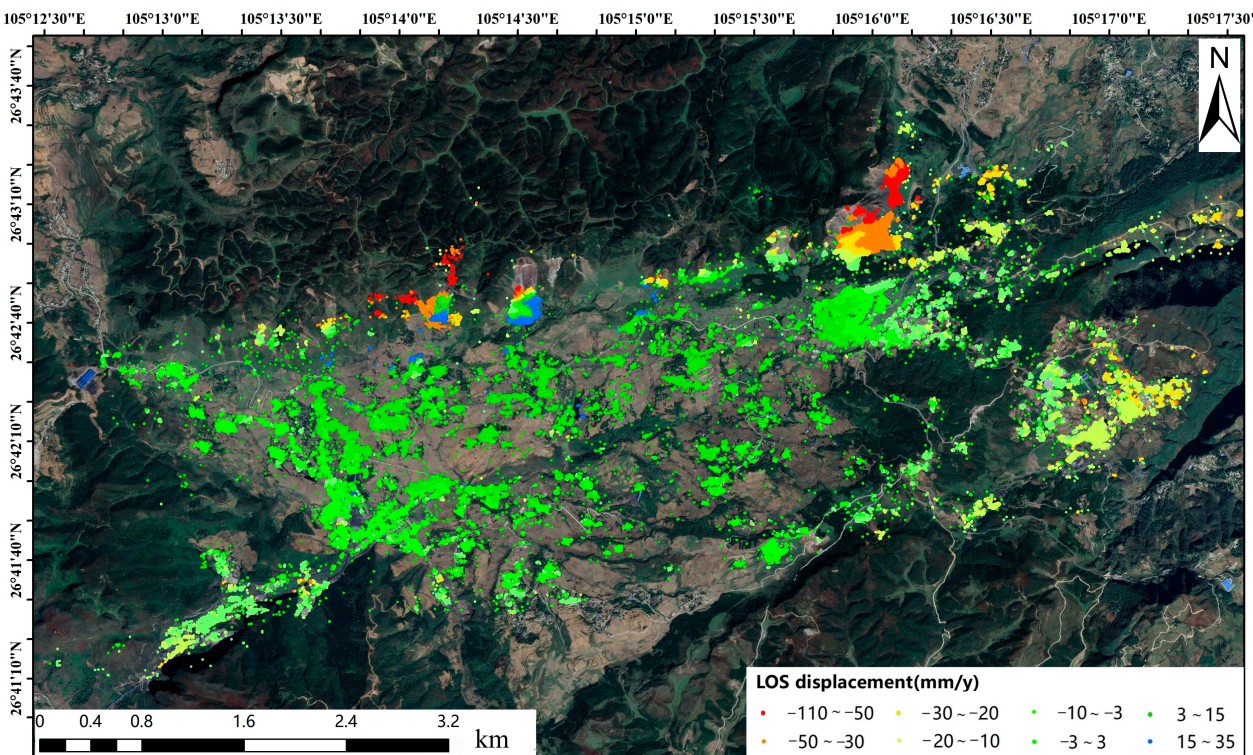

**Figure 6.** Post-processing deformation rate map of the study area.

## 5. Discussion

### *5.1. Analysis of Robust Estimation Results*

In SAR interferometry images, robust estimation of the DS points covariance matrix is realized by assigning different weights based on the extraction of homogenous pixels after the KS test. The robust estimation result can be evaluated from two perspectives: temporal coherence and the interferometric phase after phase optimization.

### 5.1.1. Temporal Coherence

The effect of robust estimation was quantitatively evaluated by the temporal coherence after phase optimization. As shown in Figure 7, the average temporal coherence of the covariance matrix without robust estimation after phase optimization was 0.423, and the standard deviation was 0.078. The average temporal coherence of the covariance matrix after robust estimation and phase optimization increased to 0.444, and the standard deviation increased to 0.084. It can be seen that the phase accuracy improved in the temporal dimension after the robust estimation of the covariance matrix.

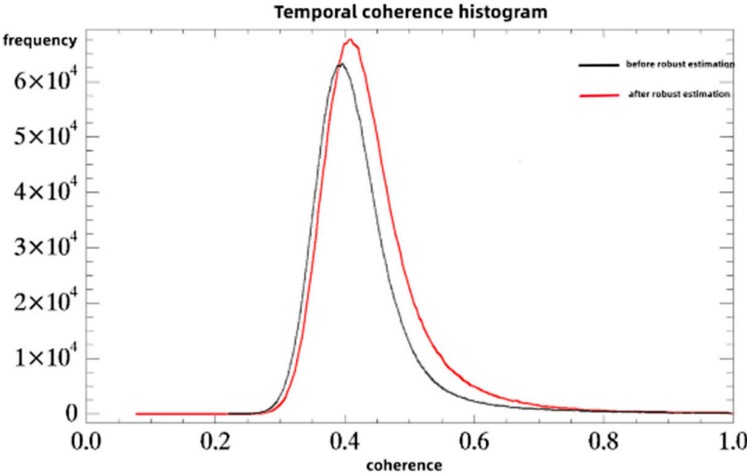

**Figure 7.** Temporal coherence histogram.

### 5.1.2. Interferometry Phase

The quantitative performance of the interferometric phase can be evaluated according to the sum of phase difference (SPD). The smaller the SPD value, the better the interferogram phase quality [44]. The average SPD before and after robust estimation is presented in Figure 8. It can be seen from the histogram that the SPD value after robust estimation was lower (red bar), indicating that the spatial phase became smoother after robust estimation. Meanwhile, the amplitude variation of SPD before and after robust estimation was calculated. The amplitude variation values were all negative. The SPD value after robust estimation decreased by at least 2.5%, compared to that without robust estimation. The mean and variance of SPD after robust estimation were lower than those before robust estimation. Thus, the effectiveness of robust estimation of the covariance matrix was verified in the spatial dimension.

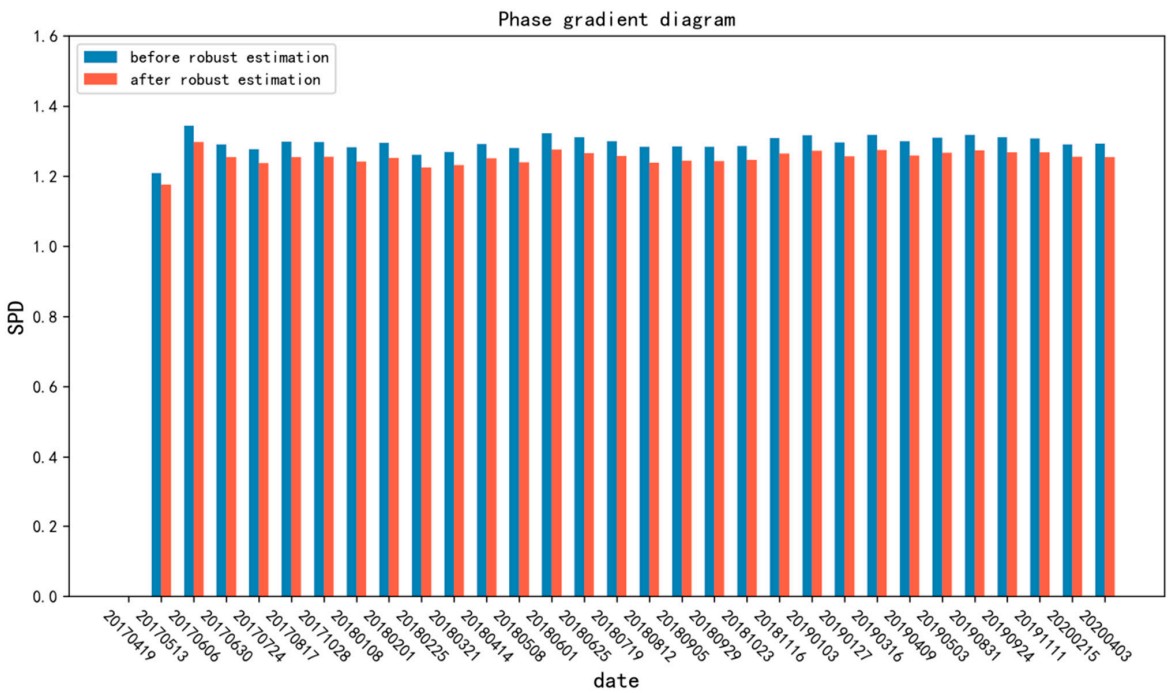

(**a**) The sum of phase difference

**Figure 8.** *Cont.*

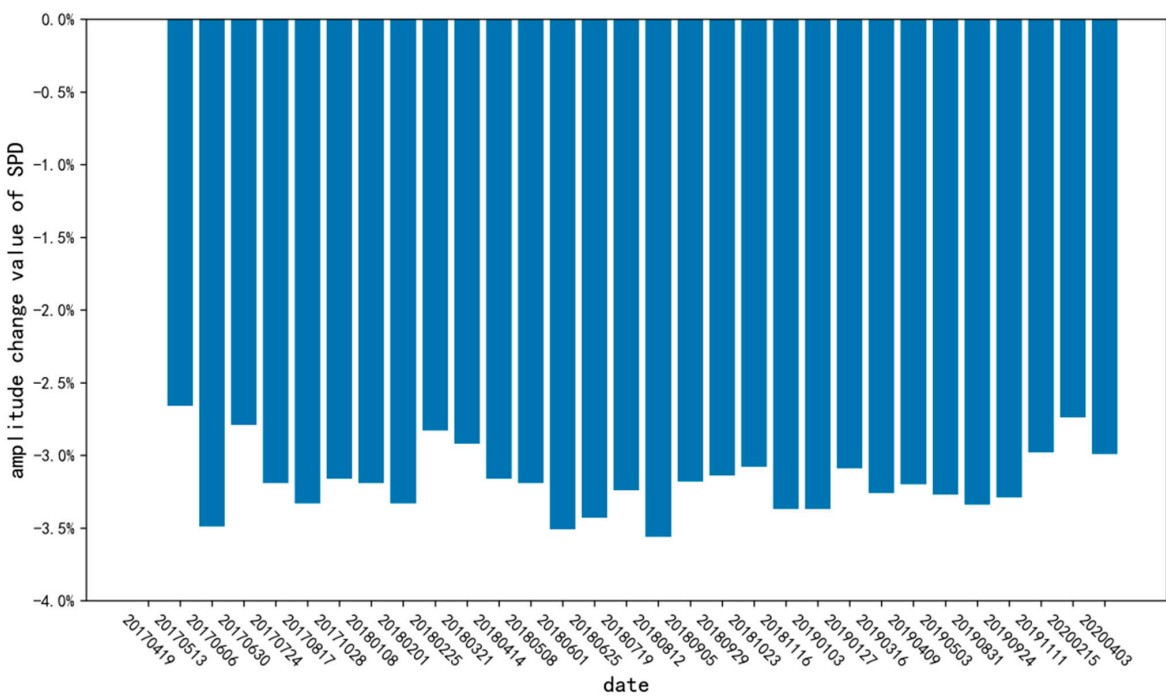

(**b**) SPD amplitude variation value

**Figure 8.** Phase evaluation of interferometry.

### 5.2. Adaptive Network Analysis

The adaptive network construction method proposed in this paper considers the number of edges, coherence, connectivity, spatial distribution, and other factors. This method was quantitatively compared with the following three methods: k-nearest neighbor network construction, range threshold network construction, and Delaunay triangulation network construction.

#### 5.2.1. Single Point Analysis

To illustrate the details of adaptive network construction, one candidate measurement point is selected from the study area to compare the spatial distribution under different network construction methods (as shown in Figure 9). For experiments, the selected candidate measurement point is located in the red box in Figure 9a. Figure 9b–e shows the spatial distribution of the point connected edges with four network construction methods: k-nearest neighbor network construction (KNN), range threshold network construction (range), Delaunay triangulation network construction (Delaunay), and adaptive network construction (adaptive), respectively. It is worth noting that, because the connected edges in Figure 9e are short, the amplification scale was compared to that of Figure 9b–d. The blue edges are the new edges added by the adaptive network construction network based on robust Delaunay triangulation network shown in Figure 9e.

The results of four network construction methods were superimposed on a radar map to quantify their advantages and disadvantages. We assigned all edges into eight different quadrants, according to the orientation angle of each edge. As shown in Figure 10, the Delaunay method had no edge in the eighth quadrant. The same result was obtained by the range threshold method, but more redundant edges appeared in other quadrants, e.g., the number of edges that belong to the fifth quadrant reached as high as 581, which was tens of times more than in other quadrants. Similarly, the KNN method missed edges in the first and fourth quadrants. Only the adaptive network method could ensure that there were edges in each region, and the number of edges was moderate. To sum up, compared with other methods, the adaptive network can overcome the shortcomings of edge unevenness

in space by increasing the number of edges in an adaptive manner to ensure that each direction contains edges.

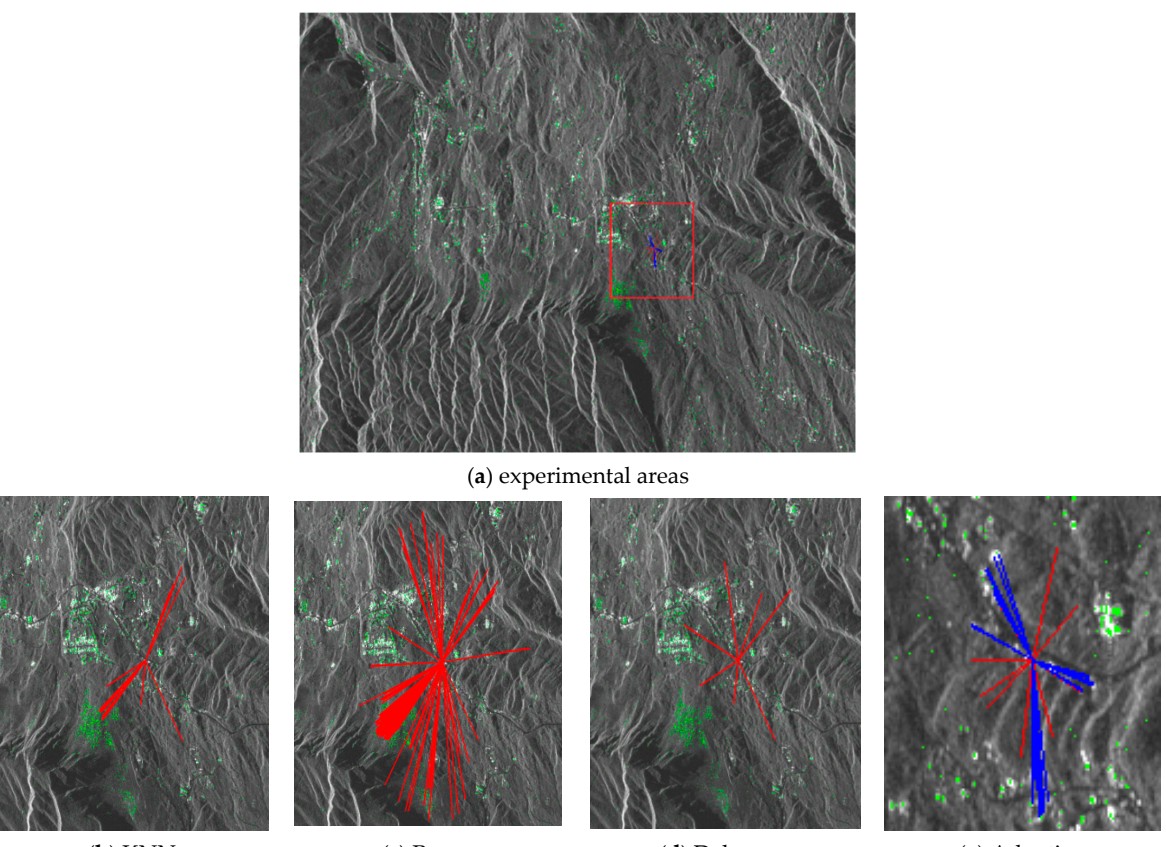

(**a**) experimental areas

(**b**) KNN   (**c**) Range   (**d**) Delaunay   (**e**) Adaptive

**Figure 9.** The spatial distribution of edges under different methods.

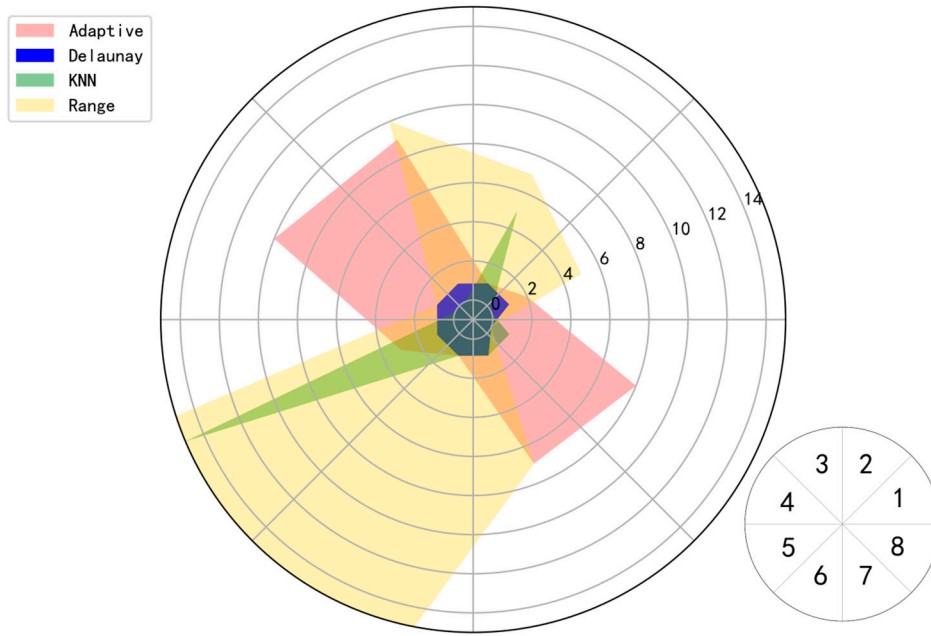

**Figure 10.** Radar statistical charts of the number of edges under different network construction methods.

### 5.2.2. Initial Network Coherence and Edge Length

In constructing the initial network, the average coherence, average edge length and the number of edges of the k-nearest neighbor network, the range threshold network, the Delaunay triangulation network, and the proposed adaptive network are presented in Table 1. It can be seen that the adaptive method proposed in this paper constructs the network through the shortest (74.946) and most coherent (0.614) edges. However, the number of edges used in the adaptive method was similar to that used in the range threshold method.

**Table 1.** Initial network parameters of different network construction methods.

| | k-Nearest Neighbor Network | Range Threshold Network | Delaunay Triangulation Network | Adaptive Network |
|---|---|---|---|---|
| average coherence | 0.492 | 0.503 | 0.451 | 0.614 |
| average edge length | 1008.72 | 325.073 | 999.63 | 74.946 |
| number of edges | 25,461 | 323,341 | 6043 | 367,652 |
| spatial distribution of highly coherent edges | non-uniform | non-uniform | non-uniform | uniform |

The coherence of the four methods is visualized in Figure 11, where the red edges represent high coherence and the blue edges represent low coherence. For the proposed method in this paper, a large number of high-coherence edges were uniformly distributed in the study area (Figure 11d), while the high-coherence edges of the other three methods were only distributed in some areas of the study area.

### 5.2.3. Skeleton Network Coherence and Edge Length

Based on the initial network, the coherence threshold was set to 0.5, and the distance threshold was set to 1200. The edges with low coherence and long distance were removed to obtain the high-quality skeleton network. Through this processing, the number of edges was reduced from 25,461 to 11,505 for the k-nearest network, and that was reduced from 6043 to 2873 for the Delaunay triangulation. This indicates that the utilization of edges was lower than 0.5. As for the network constructed with a range network and the adaptive optimization network, the edge utilization was higher, reaching 0.58. Importantly, only the adaptive method retained the spatial connectivity, and the maximum edge length decreased from 6186.88 m to 999.843 m, while the average coherence increased from 0.614 to 0.75. The detailed results are presented in Table 2.

**Table 2.** Skeleton network parameters of different network construction methods.

| | k-Nearest Neighbor Network | Range Threshold Network | Delaunay Triangulation Network | Adaptive Network |
|---|---|---|---|---|
| edges number of the initial network | 25,461 | 323,341 | 6043 | 367,652 |
| edges number of skeleton network | 11,505 | 191,814 | 2873 | 214,113 |
| utilization of edges | 0.45 | 0.59 | 0.47 | 0.58 |
| spatial connectivity of highly coherent edges | disconnected | disconnected | disconnected | connected |

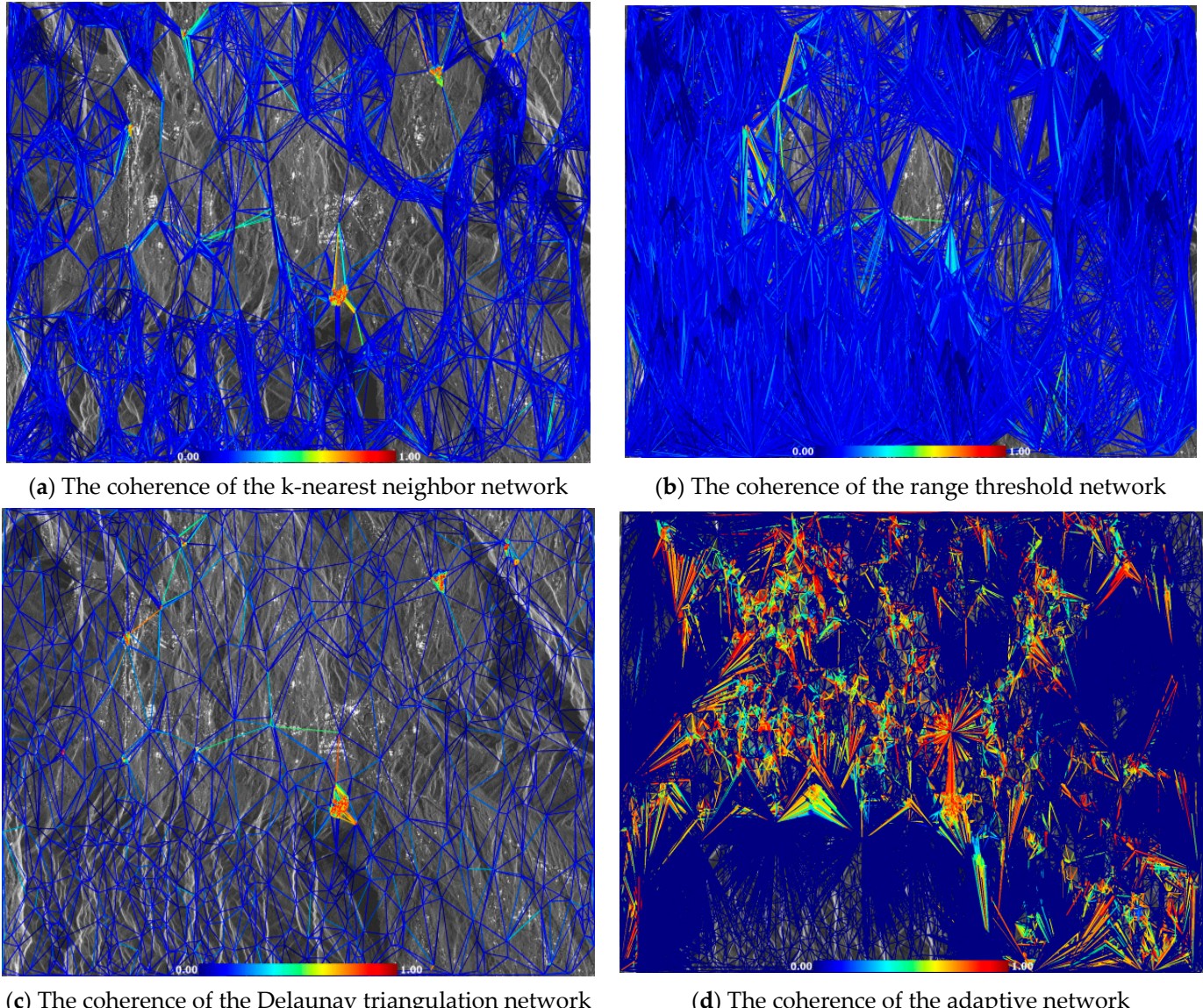

(**a**) The coherence of the k-nearest neighbor network

(**b**) The coherence of the range threshold network

(**c**) The coherence of the Delaunay triangulation network

(**d**) The coherence of the adaptive network

**Figure 11.** The coherence of the initial network under different network construction methods.

Meanwhile, the skeleton network coherence of the k-nearest neighbor network, the range threshold network, the Delaunay triangulation network, and the adaptive network were all visualized. As shown in Figure 12, only the adaptive network had a large number of highly coherent edges to maintain connectivity, and the edges were evenly distributed in the study area, while the other three methods could not guarantee the connectivity of the network. These results prove that the adaptive network maintains good connectivity with high coherence edges.

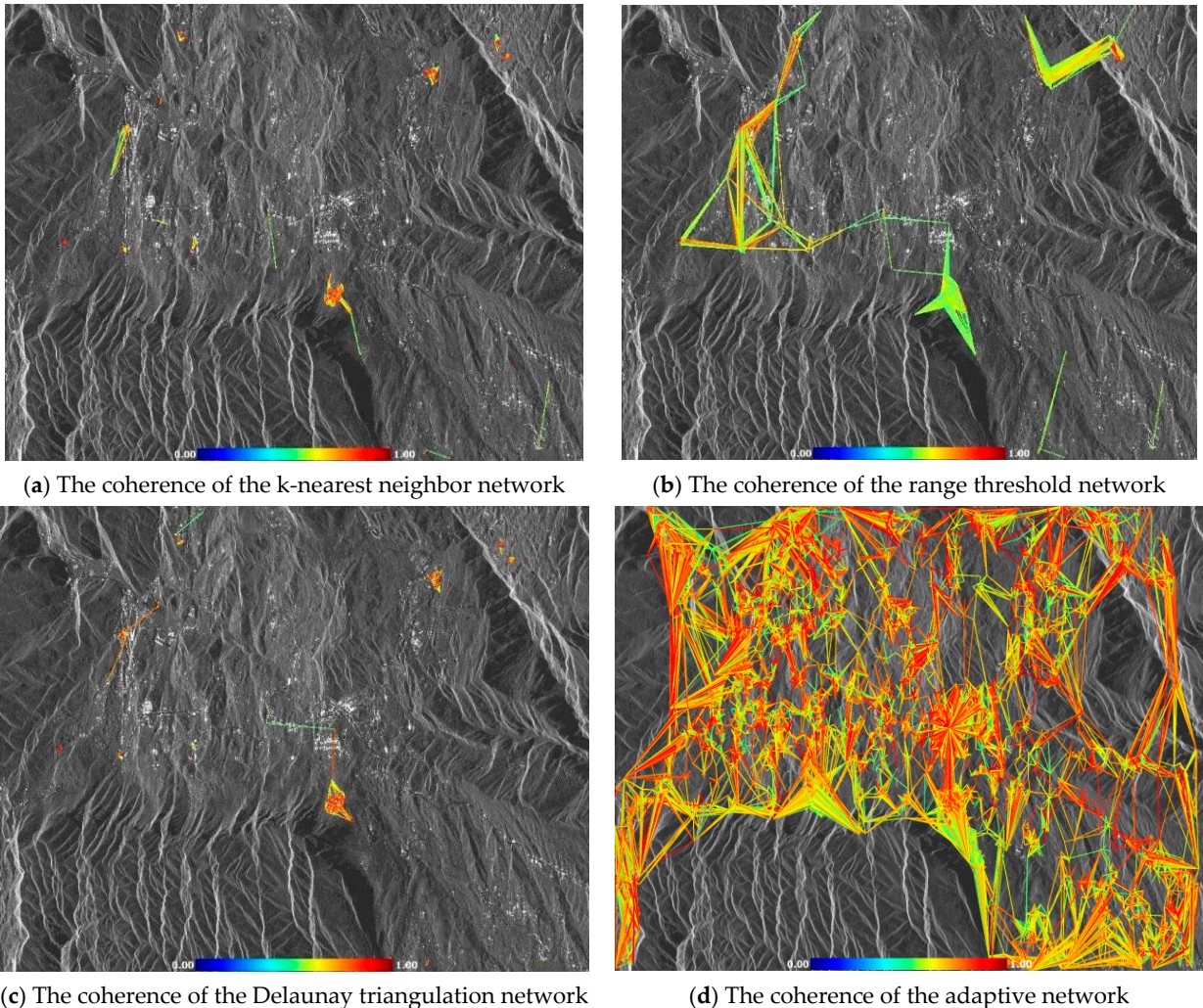

(**a**) The coherence of the k-nearest neighbor network

(**b**) The coherence of the range threshold network

(**c**) The coherence of the Delaunay triangulation network

(**d**) The coherence of the adaptive network

**Figure 12.** The coherence of the skeleton network under different network construction methods.

*5.3. Final Result of Zongling Landslide Group*

In this section, the performance of the proposed approach is evaluated, in terms of the number of pixels calculated and the detail of the generated deformation velocity maps.

To demonstrate the effectiveness of the robust estimation for DS point selection, the DS points selected by robust estimation were compared with that of non-robust estimation in the same situation, using the adaptive construction network method. As shown in Figure 13a,c, the results obtained by applying the proposed adaptive network were compared.

We compared the two methods of point selection under the frame of the adaptive network. As the deformation inversion processing eliminated some of those that were originally selected and did not pass the quality tests, robust estimation method obtains a larger number of points (78,840), compared with non-robust estimation (63,785). Consequently, robust estimation can greatly increase the number of DS points.

Moreover, the Zuojiaying landslide in the Zongling landslide group was selected for detailed description to demonstrate the effectiveness of robust estimation point selection. The deformation velocity maps of the Zuojiaying landslide, estimated by difference approaches, are shown in Figure 14. Compared with Figure 14a,b, although both PS and DS were used to select points, the robust estimation method for selecting DS points could retain more MPs, fully showing the trend of upper edge subsidence (the red part in Figure 14b) and the lower edge uplift (the blue part in Figure 14b) of the landslide. The lack of robust estimation of DS points only shows the uplift portion of the landslide's lower edge (the blue part in Figure 14a).

In addition, to demonstrate the effectiveness of the adaptive construction network, based on interferometric point target analysis (IPTA) [40], the conventional IPTA network was taken to compare with the adaptive network by adopting the same point selection mode. As shown in Figure 13b,c, the IPTA network applied to the PS and robust DS points did not obtain results that were as good as that of the adaptive network. The IPTA network only calculated 10,101 points from the total number of 130,104 points, which only accounted for 7.8% of all points. However, the adaptive network proposed in this paper could obtain 7.8 times MPs than that of the IPTA network. The validity of the adaptive network was proven by comparing the IPTA network and adaptive network.

Moreover, the Zhongling landslide with the greatest deformation in the study area was selected to show the effectiveness of adaptive network construction in detail. As shown in Figure 15a, only a partial deformation velocity map of landslide bodies could be obtained by the IPTA network construction method; therefore, it could not reflect the outline of the whole landslide. The opposite is that seen in Figure 15b, obtained by the adaptive network construction, which shows more details of landslide edges.

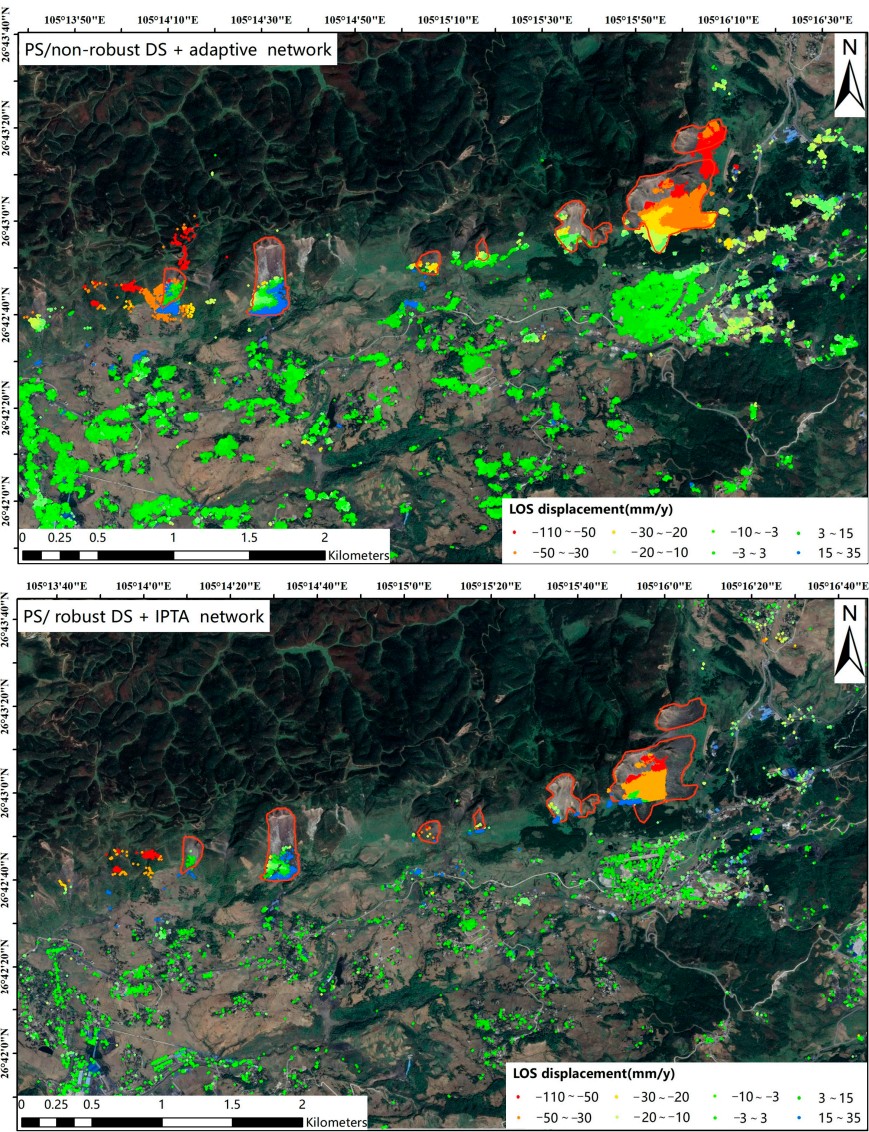

**Figure 13.** *Cont.*

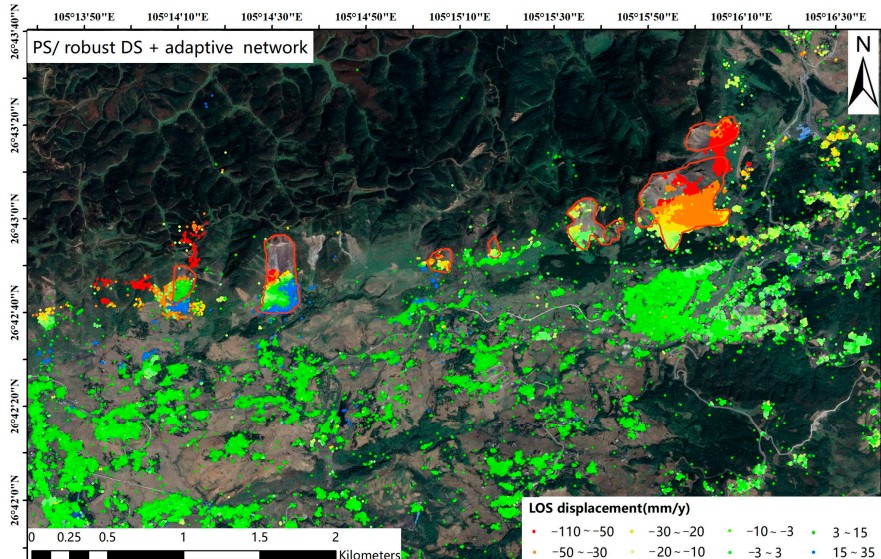

**Figure 13.** The results of deformation rate by different methods in the study area.

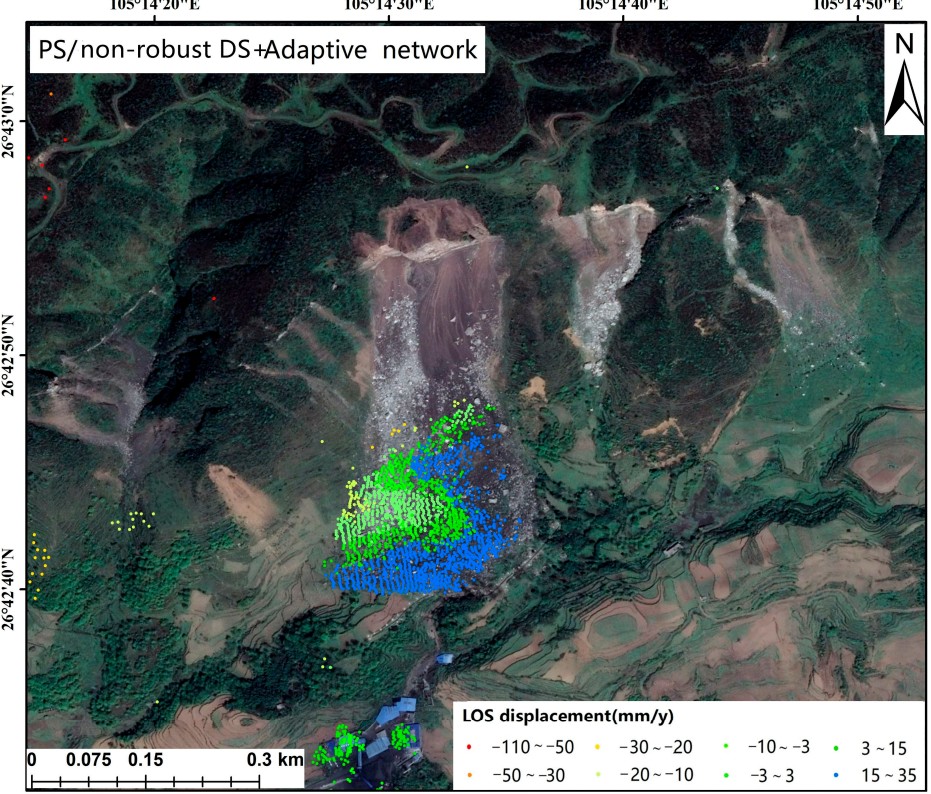

**Figure 14.** *Cont*.

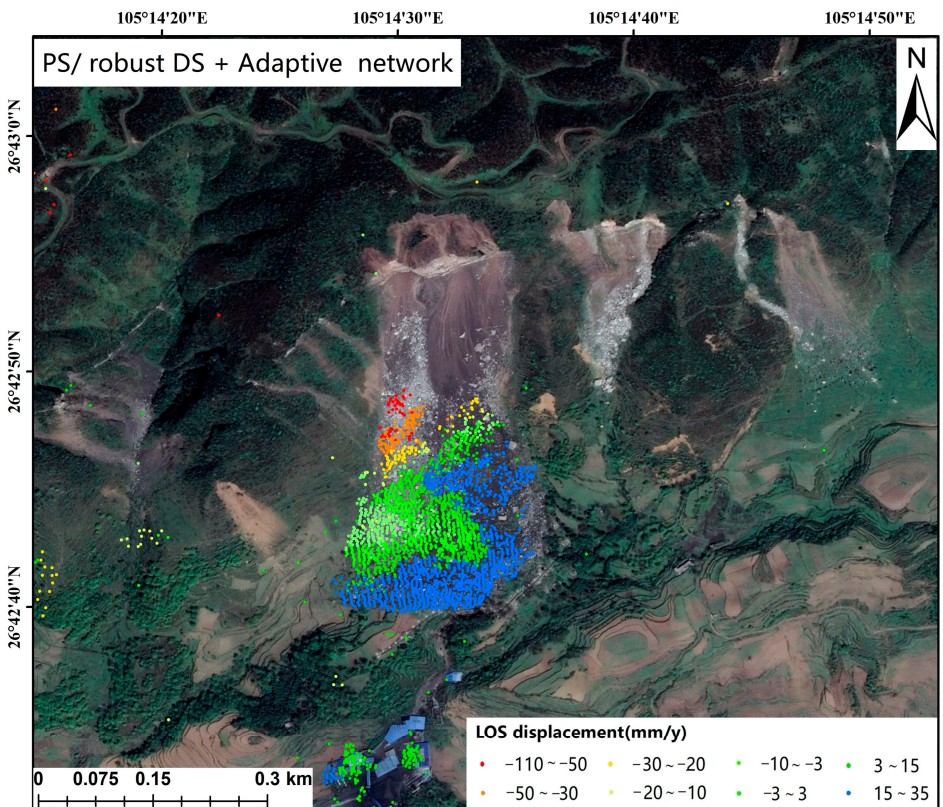

**Figure 14.** Comparison of different methods for Zuojiaying landslide.

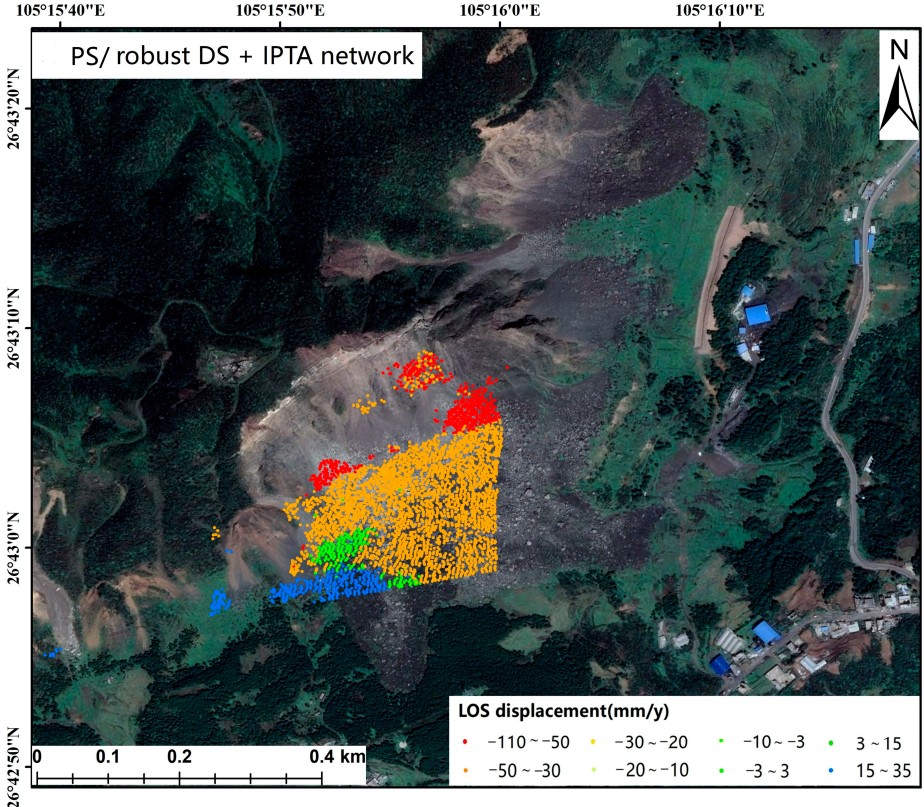

**Figure 15.** *Cont.*

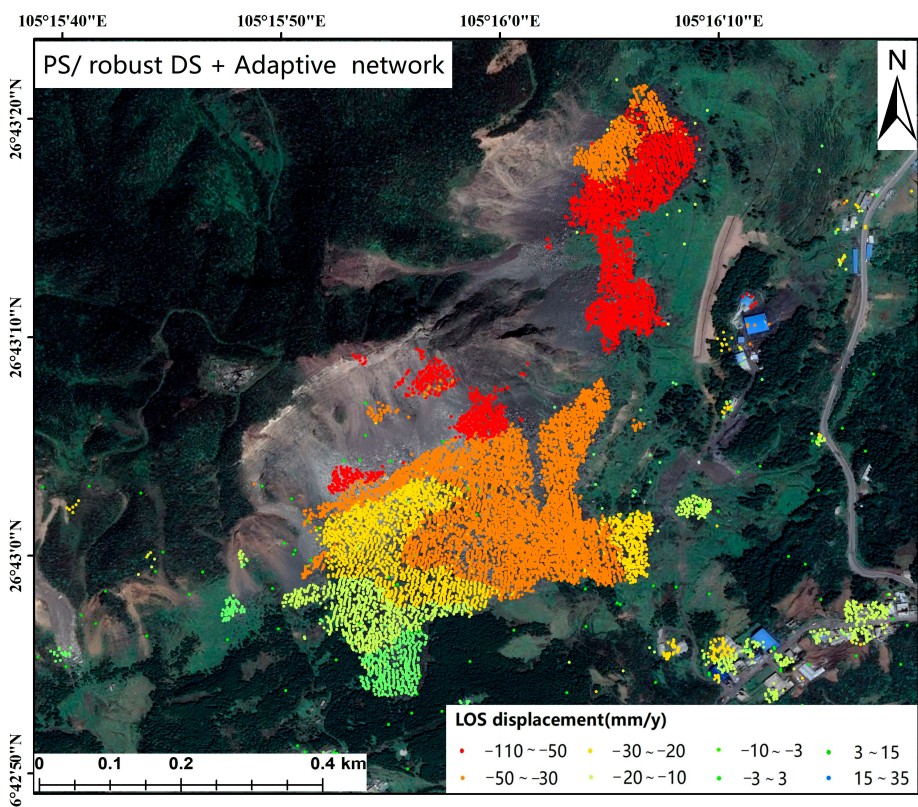

**Figure 15.** Comparison of different methods for the Zhongling landslide.

## 6. Conclusions

Due to the influence of atmospheric phase delays and terrain fluctuation in complex mountainous areas, traditional PS-InSAR technology often fails to select enough measurement points (MPs) and loses effective MPs during phase unwrapping. To solve this problem, an improved adaptive network construction algorithm is proposed in this paper. First, the covariance matrix of the DS point was estimated robustly, and then PS and DS points were combined to increase point density. By comparing the temporal coherence in the time dimension and interferometric phase in the space dimension, before and after robust estimation, it can be seen that the average temporal coherence increased by 0.02, and the SPD value of the interferometric phase decreased by at least 2.5%. Then, based on the traditional Delaunay triangulation network, the adaptive network was constructed by considering the edge length, edge coherence, edge number, connectivity, and spatial distribution. It was demonstrated that the adaptive network construction method was superior to the k-nearest neighbor network, range threshold network, and Delaunay triangulation network. Finally, the performance of the proposed approach was evaluated in terms of the number of pixels calculated and the detail of the generated deformation velocity maps by using 31 RADARSAT-2 images covering the Zongling landslide group in Guizhou Province. The robust DS points were applied to invert the deformation information, in which 15,055 more points could be obtained, compared with non-robust estimation for selecting DS points. The adaptive network was applied to invert the deformation information, which could obtain 7.8 times more points and reveal more details about the landslide than the IPTA network. The proposed algorithm improved the number of effective MPs and accuracy of phase unwrapping.

## 7. Patents

Adaptive network construction methods, devices, and equipment. (CN202210741068.X).

**Author Contributions:** Conceptualization, B.T. and C.X.; methodology, Y.Z.; software, B.T.; validation, Y.Z., Y.Y. and H.F.; investigation, M.Z.; data curation, Y.Z.; writing—original draft preparation, Y.Z.; writing—review and editing, B.T. and C.X.; visualization, Y.Z.; supervision, Y.G. and C.S.; project administration, Q.W. and R.W. All authors have read and agreed to the published version of the manuscript.

**Funding:** This research was funded by National Key R&D Program of China (2022YFC3005601), and InSAR geological hazard surface deformation monitoring project (E0H2111902).

**Institutional Review Board Statement:** Not applicable.

**Informed Consent Statement:** Not applicable.

**Data Availability Statement:** Not applicable.

**Acknowledgments:** The Third Institute of Surveying and Mapping of Guizhou Province.

**Conflicts of Interest:** The authors declare no conflict of interest.

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
