# Peer review of "Multi-Temporal InSAR Deformation Monitoring Zongling Landslide Group in Guizhou Province Based on the Adaptive Network Method"

_sustainability, doi:10.3390/su15020894_

Round 1
Reviewer 1 Report (New Reviewer)
Dear Author,
The manuscript submitted for evaluation is very interesting and addresses a topic of relevant interest to the scientific community. The theoretical framework is based on updated references. It is very clear in terms of the employed methodology, which seems to be appropriate, and reveals a clear and logical structure. The text is clear and properly drafted. The figures seem to be appropriate. The research and the results obtained are important, I consider the contents of this paper to be of good quality and could be published in it’s current format.
Author Response
Thank you for your review and the positive comments.
Reviewer 2 Report (New Reviewer)
In this paper, an adaptive construction algorithm is used to improve the number of MPs and accuracy of the phase unwrapping. By estimating the covariance matrix of the DS points, the weights of the heterogeneous samples are reduced to improve the coherence, which ultimately increases the density of MP points. And adaptive enhancement of traditional Delaunay triangulation networks considering edge length, edge coherence, number of edges, connectivity and spatial distribution. There are a number of issues listed below which need to be addressed.
1.P3L107: suggest: add ‘.1)’ or delete ‘.) 2’
2.P4L141: suggest: Inside Fig.1(b) add the label of Distribution of Mineral. It is not immediately clear that the grey rectangles labels could be associated with the corresponding area.
3.P6L173: comment: What does the superscript H in Equation 1 stand for? Authors should properly check the expressions of other Equations below, e.g. is the accent symbol missing from I in Equation 9?
4.P8L280: The paper mentions that the coherence threshold is set to 0.65 to ensure the connectivity of the time-series, but why is there a time baseline in the Fig.3 where the coherence is less than 0.65. In addition, authors may need to modify the x-axis labels in Figure 3 below.
5.P9L318: When expressing the advantages of the improved method in this paper here, statistics on the number of edges or the number of MPs can be added.
6.P11L329: suggest: It may be better to unify the colorbar in (a),(c),(e),(g) in Fig. 5 and place them in a better position. Because now looking at the bottom of the figure one of the long edges is obscured.
7.P12L361: why after robust estimation, the standard deviation is increased? What are the main possible reasons?
8.P16L420: suggest: Figure 10 could be improved by aligning the angle of the subplot expressing the quadrant partition with the radar plot and setting the transparency appropriately.
9.P17L446: The authors mentioned earlier that the distance threshold is set to 1200m, but why is it 1000m here?
Author Response
Thank you for your kind and careful reviewing. See Annex for responds to the reviewers’ comments.

This manuscript is a resubmission of an earlier submission. The following is a list of the peer review reports and author responses from that submission.
Round 1
Reviewer 1 Report
see attached file

Reviewer 2 Report
Dear Authors
Honestly, it was quite embarrassing to review the disorganized, poorly written draft with confusing words and sentences. It seems that the drafting clearly skipped cross-checks by the authors regardless of scientific achievements, which may be included in the draft.
Too many mistakes in writing and lack of liability indicate this draft was written in haste.
I can't point all but roughly as follows
1) L13 atmospheric path delay: It's not a significant factor in PS application on landslide areas. Refer to references coherence loss on slope side and explain
2) L69-L111 : This part is very difficult to understand, not because of the baseline technical issue but improper writing. Do not deliver anything.
3) Fig 2. Not readable
4) L182 : Is such a sub-sectional method allowable?
5) L193-196 Check writing
6) L916 where is section 2.2? Is it section 3.2? Why does this content precede the detailed description in section 3.2
7) L217 "a"
8) L225 From here, all descriptions of formulas do not follow a conventional way and are completely not readable. Rewrite/restructure all involved texts and redefine equations
8) L241 Same with 6. Such sectioning is absurd.
9 ) L235 unnatural writing - so many such cases
10) L260 : Where was AD used in this study?
11) Section 3.2 this part has better writing than section 3.1 but the descriptions of equations are still bad.
12) L316 : what is the non-robust method?
13) L351 : Check wording
14) L357, L359 : Please describe the equations properly
15) L411-424 : No connection to other technical introductory. The expression of radarmap is not described.
16) L449 : How was SPD driven? What are the significances of differences in Fig 11 to 13?
17) Table 1 2 the last row is not necessary if those were stated in the text.
18) Table 3 is not necessary. Rather, IPTA needs to be described enough for comparison in the front section.
19) Figure 15-19. I can't see any clues to prove the strengths of the proposed methods compared to other cases.
20) How can the author state the upgrade of 0.02 temporal coh and SPD 2.5% are evidence of their technical improvements? There is absolutely no validation.
References are so poor.
Best Regards